# CAST as a Potential Oncogene, Identified by Machine Search, in Gastric Cancer Infiltrated with Macrophages and Associated with Lgr5

**DOI:** 10.3390/biom12050670

**Published:** 2022-05-06

**Authors:** Kuang-Tsu Yang, Chia-Chi Yen, Renin Chang, Jui-Tzu Wang, Jin-Shuen Chen

**Affiliations:** 1Division of Gastroenterology and Hepatology, Department of Internal Medicine, Kaohsiung Municipal Min-Sheng Hospital, Kaohsiung 802213, Taiwan; ktyang1104@gmail.com; 2Division of Family Medicine, Department of Community Medicine, Kaohsiung Municipal Min-Sheng Hospital, Kaohsiung 802213, Taiwan; 3Division of Gastroenterology and Hepatology, Department of Internal Medicine, Kaohsiung Veterans General Hospital, Kaohsiung 813414, Taiwan; 4Institute of Biomedical Sciences, National Sun Yat-Sen University, Kaohsiung 804201, Taiwan; 5School of Medicine, College of Medicine, National Taiwan University, Taipei 10051, Taiwan; 6Superintendent’s Office, Kaohsiung Municipal Min-Sheng Hospital, Kaohsiung 804201, Taiwan; ycc.lun@msa.hinet.net; 7Department of Nutrition, Institute of Biomedical Nutrition, Hung-Kuang University, Taichung 433304, Taiwan; 8Department of Business Management, National Sun Yat-Sen University, Kaohsiung 804201, Taiwan; 9Department of Medical Education and Research, Kaohsiung Veterans General Hospital, Kaohsiung 813414, Taiwan; rhapsody1881@gmail.com (R.C.); tsz882021@gmail.com (J.-T.W.); 10Department of Emergency Medicine, Kaohsiung Veterans General Hospital, Kaohsiung 813414, Taiwan; 11Institute of Biotechnology and Chemical Engineering, I-Shou University, Kaohsiung 84001, Taiwan; 12Department of Recreation and Sports Management, Tajen University, Pingtung 90741, Taiwan; 13Division of Nephrology, Department of Internal Medicine, Kaohsiung Veterans General Hospital, Kaohsiung 813414, Taiwan; 14Faculty of Medicine, School of Medicine, National Defense Medicine, Taipei 11490, Taiwan

**Keywords:** CAST, Lgr5, WNT, gastric cancer, machine searching, macrophage

## Abstract

Background: Gastric cancer (GC) is one of the leading malignant diseases worldwide, especially in Asia. CAST is a potential oncogene in GC carcinogenesis. The character of macrophage infiltration in the GC microenvironment also remains unaddressed. Methods: We first applied machine searching to evaluate gene candidates for GC. CAST expression and pan-cancer surveyance were analyzed using the Human Protein Atlas (HPA) and Gene Expression Profiling Interactive Analysis 2 (GEPIA2) database. The protein–protein interaction (PPI) network was downloaded from STRING. We investigated the impact of CAST on clinical prognosis using a Kaplan–Meier plotter. The correlations between CAST and Lgr5 and macrophage infiltration in GC were determined using TIMER 2.0. Finally, GeneMANIA was also used to evaluate the possible functional linkages between genes. Results: After the machine-assisted search, CAST expression was found to significantly influence the overall survival of GC patients. STRING revealed CAST-related proteomic and transcriptomic associations, mainly concerning the CAPN family. Moreover, CAST significantly impacts the prognosis of GC based on the validation of other datasets. Notably, high CAST expression was correlated with worse overall survival in GC patients (hazard ratio = 1.59; log-rank P = 9.4 × 10^−8^). CAST and Lgr5 expression were both positively correlated with WNT 2 and WNT 2B. Among the GC patients in several datasets, CAST and macrophage infiltration, evaluated together, showed no obvious association with poor clinical overall survival. Conclusions: CAST plays an important role in the clinical prognosis of GC and is associated with WNT 2/WNT 2B/Lgr5. Our study demonstrates that CAST’s influence on overall survival in GC is regulated by macrophage infiltration.

## 1. Introduction

Gastric cancer (GC) is one of the most prolific diseases worldwide. It has been estimated that there are more than 1 million newly diagnosed GC patients worldwide each year. GC is the fourth most common cancer and the second most common cause of death worldwide [1]. Globally, one in 33 men and one in 78 women will develop GC in their lifetimes [2,3]. Since GC is often diagnosed at an advanced stage, the mortality rate is high. In 2018, 784,000 people died from GC worldwide, twice as many men as women, with East Asia, Eastern Europe, and South America being the regions with the greatest GC incidence and deaths [4]. Clinically, we can expect to see more cases of GC in the future due to the aging of the population. In recent years, we have even observed an increase in the incidence of GC in young people [5].

Approximately 10% of GC patients have familial genetic clusters, and approximately 1–3% of them have mutations [6]. Familial GC includes at least three major classifications: hereditary diffuse GC (HDGC), gastric adenocarcinoma and proximal gastric polyps and disease, and familial gastrointestinal cancers [7,8,9]. To explore the frontier of the mechanisms of gastric carcinogenesis, recent studies have considered Lgr5 as an activator of the WNT signaling pathway, which promotes the proliferation of gastric adenocarcinoma cells. Stem cells overexpressing the marker Lgr5 are derived from the stomach, kidneys, colon, hair follicles, and mammary glands [10]. Wu et al. found Lgr5 expression at the bottom of normal gastric gland units and revealed differential expression in GC with varying differentiation. Furthermore, Lgr5 and Bmi1 were identified as marking the same stem-cell population. CD133, CD26, CD44, and ALDH1 associated with Lgr5 may be related to the growth of GCs [11].

Calpastatin (CAST) is usually found in the plasma membrane and surrounding the nucleus [12]. CAST inhibits calpains, which can translocate into the nucleus and further regulate the WNT/β-catenin pathway [13]. The single CAST gene can encode eight or more CAST polypeptides, ranging from 17 to 85 kDa in molecular weight, with the functions of binding to calpain molecules and Ca^2+^ dependency. The CAST/calpain system regulates a variety of cellular processes, involving the remodeling of cytoskeletal/membrane attachments, multi-signal transduction pathways, and cell apoptosis. The CAST/calpain system also participates in numerous membrane-fusion events, such as neural vesicle exocytosis and platelet aggregation [14]. CAST has previously been reported as a possible novel marker in GC development. Liu’s study results revealed that calpastatin levels were decreased in GCs. Furthermore, the ratio of (calpain 1 (CAPN1) × calpain 2 (CAPN2))/(calpastatin × calmodulin (CaM)) has been considered a potential index for GC diagnosis [15].

In recent years, tumor-associated macrophages (TAMs) have been associated with the tumor microenvironment, acting in both tumor-promoting and tumor-suppressing manners [16]. TAMs are categorized into the anti-tumor M1 phenotype (classically activated state) and the protumorigenic M2 phenotype (alternatively activated state), reflecting the Th1–Th2 polarization of T cells [17]. TAMs participate in innate host defenses and kill tumor cells. Meanwhile, TAMs also play a critical regulatory role in epithelial–mesenchymal transition, angiogenesis, and immunosuppression, hampering the efficacy of chemotherapy [18,19].

However, the characteristics of CAST associated with the immunological responses of macrophages and their relevance to Lgr5 remain unaddressed. We aimed to explore the possible interactions of the above-mentioned characteristics.

## 2. Materials and Methods

### 2.1. The Cancer Genome Atlas (TCGA) Program Analysis Using Machine Searching

The expression levels for the CAST gene in various types of cancers were identified in the Human Protein Atlas (HPA) database (https://www.proteinatlas.org/, accessed on 1 September 2021). We used Python Selenium (Version 3.8) to automatically search the TCGA database by entering different gene candidates, and we recorded all the candidate genes associated with the overall survival (OS) rate for GC. Then, the most relevant genes, including CAST and WNT (*p*-value < 0.001), were precisely selected.

### 2.2. Protein–Protein Interaction (PPI) Network from STRING

The STRING database (version 11.5) [20] is applied in the search for PPIs that are of interest to scientists and worthy of investigation. Proteins relevant to the same topic can be linked by direct and indirect relationships and mapped to a weight network in STRING, containing 14,094 organisms, 67.6 million proteins, and >20 billion interactions. Proteins are marked as nodes, and every two proteins is given as an edge and highlighted with a confidence score. The higher the confidence score, the greater the number of analogous functions among proteins [21].

### 2.3. CAST Bioinformatics Analysis Using Gene Expression Profiling Interactive Analysis 2 (GEPIA2) Datasets

We examined the mRNA levels for CAST, comparing tumor and matched normal samples using the GEPIA2 database, which can provide cancer genomic data on the basis of TCGA, and the GTEx [22].

### 2.4. Using Human Protein Atlas (HPA) for Further Validation of CAST in Different Human Tissues

We used the HPA, which is one of the most robust and comprehensive databases of protein and RNA in tissues and cells. The HPA’s goal in the Cell Atlas is to map the subcellular distributions of all human proteins over the course of a cell cycle in a canonical human cell. The HPA includes over 85% of all human protein-coding gene data. Furthermore, both immunohistochemistry (IHC) scoring parameters and subcellular localization classifications are purified to increase the numbers of cell types and organelles, and supply clinicians with bioinformatic information on intraorganellar locations. The HPA can contribute to deeper investigations for both basic and clinical research [23]. We used transcriptomic and proteomic expression to represent the characteristics of CAST in different tumor tissues Appendix A.

### 2.5. Survival Analysis Using Kaplan–Meier (KM) Plotter

The cancer-survival information and CAST bioinformatic information for the GC patients contained in the KM plotter database were extracted from the Gene Expression Omnibus (GEO), the Cancer Biomedical Informatics Grid, and The Cancer Genome Atlas database. The following GC datasets were retrieved from the GEO database: GSE62254, GSE22377, GSE51105, GSE14210, GSE29272, and GSE15459 (https://kmplot.com/analysis/index.php?p=service&cancer=gastric, accessed on 1 September 2021) [24]. We also acquired KM survival plots, in which the numbers of cancer patients for a specific period are compared between subgroups with different gene-expression statuses. We determined the hazard ratios (HRs), 95% confidence intervals (CIs), and log-rank *p*-values. A *p*-value < 0.05 was considered statistically significant.

### 2.6. TIMER 2.0 Database for Genes and Infiltrating Immune Cells

The TIMER 2.0 database (http://timer.cistrome.org/, accessed on 1 September 2021) is a website that sources a large amount of immune and gene bioinformatic information, which can be used to further analyze and summarize tumor immune-infiltration scores, such as for neutrophils, macrophages, T cells, B cells, and NK cells. TIMER 2.0 can also analyze specific oncogene mutation groups, and genes have been input for the analysis of well-known oncogenic mutations in specific tumors [25,26,27]. The correlations among the Lgr family, CAST, WNT family, and macrophages were surveyed, the data for which were taken from the TCGA database. The results of the surveyance were downloaded to observe the outcome. The relationship between the CAST gene and well-known immune infiltration in tumors was also analyzed using TIMER 2.0 for confirmation. A *p*-value < 0.05 was considered statistically significant.

### 2.7. Gene and Protein Networks Analysis

GeneMANIA (http://genemania.org/, accessed on 15 August 2021, version 3.6.0) is a real-time multiple association network integration algorithm for predicting gene function [28]. The data could be extracted for gene–gene interactions (GGIs) in our study. Regarding previous studies concerning the WNT family related to gastric cancer development, we surveyed the relationships among the WNT, CAST, and Lgr5 genes. Moreover, we analyzed the functions involving G-protein-coupled receptor binding, the canonical WNT signaling pathway, stem-cell differentiation, and the positive regulation of the WNT signaling pathway for the demonstration of GGIs.

### 2.8. Statistical Analysis

The results from the KM plotter and TIMER 2.0 are shown with the hazard ratios (HRs) and Cox *p*-values from a log-rank test. We evaluated the correlation of gene expression using Spearman’s rank correlation and statistical significance. Rho-values were applied in the determination of positive or negative correlations in protein/RNA expression.

## 3. Results

### 3.1. CAST-Centered Network Interaction and Clustering Analysis

CAST was introduced into the STRING database to obtain the functional protein-correlation network. The PPI network of this functional protein expression relevant to CAST contained 11 nodes and 40 edges, obtained with confidence scores for CAPN2/CAPN1/CAPNS1 of 0.999/0.999/0.986. The enriched *p*-value was 1.12 × 10^−11^. The K-means algorithm for clustering analysis in the constructed network of interaction, causing three distinct numbers of interactive networks, is represented in Figure 1. In addition, the gene ontology (GO) bioinformation related to CAST is shown in Table 1.

### 3.2. CAST Expression in Different Tissues

We extracted the CAST RNA-sequencing expression level from the GEPIA2 database. Figure 2 demonstrates the CAST expression in transcripts per million (TPM). Glioblastoma (GBM), pancreatic adenocarcinoma (PAAD), and stomach adenocarcinoma (STAD) showed prominent CAST expression, while testicular-germ-cell tumors (TGCTs), uterine corpus endometrial carcinoma (UCEC), and uterine carcinosarcoma (UCS) involved less. The CAST expression in different tissues is shown in Table 1.

### 3.3. Validation of CAST Expression in GC

To gain robust confidence in the association between CAST and GC, we further mined the HPA, with the cancer types color-coded according to which types of normal organ the cancers originated from, including HPA036881, HPA036882, and CAB009491, as shown in Figure 3A–C, respectively. No patients with high expression, six patients with medium expression, three patients with low expression, and three patients with undetected expression of CAST were recorded in HPA036881. No patients with high expression, two patients with medium expression, two patients with low expression, and eight patients with undetected expression of CAST were recorded in HPA036882. Four patients with high expression, five patients with medium expression, one patient with low expression, and two patients with undetected expression of CAST were recorded in CAB009491. An overview of the RNA expression is shown in Figure 3D.

### 3.4. CAST Associated with Survival in GC

Table 2 shows significant differences in survival between the low-expression and high-expression cohorts. Among the cohorts, the low expression of CAST cohort had longer median survival than the high expression of CAST cohort, except for GSE62254. *p*-value was statistically significant (<0.05) in all, GSE22377, GSE14210, GSE29272, and GSE15459 (Table 3).

In GC-cohort analyses from the KM plotter, CAST was significantly related to patient survival (all, HR: 1.59; 95% confidence interval (CI): 1.34–1.88; log-rank *p*-value: 9.4 × 10^−8^) when the median expression of CAST was set as a cutoff point for stratifying patients in Figure 4A. In Figure 4B, most subgroups showed lower survival in the high CAST expression cohorts than in the low CAST expression cohorts with significant *p*-values.

### 3.5. External Validation for KM Plotter Using TIMER 2.0 Datasets

Figure 5 shows the relevance of CAST alone to GC survival (HR) and clinical outcome (HR: 1.22; *p* = 0.0415), which was compatible with the dataset retrieved from the KM plotter. In this analysis, 5-year survival was measured, and the low CAST expression cohort was found to have higher cumulative survival than the high CAST expression cohort with clinical significance.

### 3.6. CAST/WNT/Lgr5 Co-Expressions in GC

Figure 6 and Figure 7 show the correlation between CAST and the WNT family, and Lgr5 and the WNT family, respectively. WNT2, WNT16, WNT2B, WNT5A, WNT9A, and WNT9B showed positive correlations with CAST. On the other hand, WNT6, WNT3A, and WNT8B revealed negative correlations with CAST. Moreover, WNT2, WNT3, WNT11, WNT2B, WNT7B, WNT8B, and WNT10B had positive correlations with Lgr5. The overlapping WNT-family genes for both Lgr5 and CAST were WNT2 and WNT2B.

### 3.7. CAST and Macrophages in GC

TIMER 2.0 showed databases including TIMER, EPIC, XCELL, CIBERSORT-ABS, and QUANTISEQ. We discovered that, as shown in Figure 8, TIMER indicated that high CAST expression and high macrophage infiltration were significantly associated with lower cumulative survival than high CAST expression and low macrophage infiltration, with an HR of 2.08 and a *p*-value of 0.00927. However, there was no significance regarding cumulative survival according to the EPIC and XCELL CAST expression and macrophage-infiltration analyses. The evaluation of M1 and M2 macrophages also showed no significant difference in cumulative survival.

### 3.8. CAST–WNT2/WNT2B–Lgr5 Linkages Associated with Gastric Carcinogenesis

We input CAST, WNT2, WNT2B, and Lgr5, using GeneMANIA, and found that CAST was linked to the WNT family and Lgr family, as shown in Figure 9. WNT2 and WNT2B were linked to G-protein-coupled receptor binding. WNT2 was linked to the canonical WNT signaling pathway, but WNT2B did not involve it. Lgr5 showed positive regulation of the WNT signaling pathway and canonical WNT signaling pathway.

## 4. Discussion

In our present study, we demonstrated that CAST is an oncogene associated with Lgr5 in gastric cancer via the WNT signaling pathway. The expression of WNT2 and WNT2B showed a significant positive correlation with both CAST and Lgr5, which warrants further study of the molecular biochemistry, transcriptomics, and proteomics in GC. Though CAST has been discovered to have a prominent impact on GC patients’ survival, after multivariate adjustments, multi-database datasets revealed that macrophages might play a key role in immune regulation in the GC microenvironment, promoting tumor suppression.

Our research revealed CAST as a potential oncogene promoting GC formation. Previous studies seldom focused on this novel issue. Liu et al. [15] proposed that—other than CAST—CAPN1, CAPN2, and CaM might also contribute to GC formation, which is partially compatible with our results. The calpain system was also associated with colorectal adenocarcinoma and prostate cancer, which suggested that calpains might be important in tumor progression [29,30]. The calpain system is relevant to human epidermal growth factor receptor 2 and E-cadherin in breast cancer [31,32]. Meanwhile, calpain-2 was proven to contribute to the methylation of CRMP4′s promoter, repressing its transcription, thereby promoting the metastasis of prostate cancer by enhancing expression of vascular endothelial growth factor C [33].

The mechanism by which CAST promoted GC remained unclear. We tried to identify relevant gene expression or possible pathways. After database mining, Lgr5 and CAST were found to possibly regulate GC formation via the same pathway—the signaling of the WNT family, especially WNT 2 and WNT 2B—representing novel findings regarding the signature of GC formation. The WNT/β-catenin pathway in gastric cancer was shown to be important in regulating proliferation, stem-cell maintenance, and homeostasis in the gastric mucosa [34,35]. Activated WNT/β-catenin signaling can be observed in more than 30% of GCs. The fundamental role of WNT/β-catenin signaling in the self-renewal of GC stem cells has been demonstrated [36,37,38]. The WNT/β-catenin signaling paradox was recently discussed, with regard to the hyperactivation of WNT signaling by mutations in β-catenin destruction complex components or β-catenin itself contributing to tumorigenesis [39]. β-catenin can be further activated by additional layers of regulation, highlighting the complicated nature of the role of WNT signaling deregulation in cancer [40,41,42]. The dual function (tumorigenesis or tumor suppression) of the WNT/β-catenin system was highlighted in our clinicopathological dataset survival follow-up.

Recently, TAMs were discovered to be associated with WNT signaling in the tumor microenvironment. Wu et al. [43] demonstrated that macrophages play a protumorigenic role in GC patients. The mechanism could originate from tumor-microenvironment-related inflammation, matrix remodeling, angiogenesis, seeding at distant sites, intravasation, or tumor-cell invasion [44]. The current studies also provide scientists with a clue that macrophages may play a helpful or harmful role in the GC microenvironment. Huang et al. also demonstrated that the heterogeneity of macrophages within the tumor is present at both the macro- and microlevels due to the gradient changes in different markers [45]. In our study, the role of macrophage infiltration in GC associated with CAST remained unclear regarding GC formation and survival. We hypothesize that macrophage infiltration could manipulate specific signaling pathways in GC carcinogenesis. Perhaps further in vitro research should be conducted to determine the mechanism.

We had confidence in the database mining for genes and macrophages relevant to GC on the basis of certain characteristics, such as the high reproducibility, high convenience, and lack of need to inform and obtain consent from patients. The analytical methodology of our article is very suitable for establishing a precise/personalized evaluation of the molecular investigation of GC. Though we found a novel marker and immune infiltration to be correlated with GC, we acknowledge some limitations in our study. First, though the databases contain a large amount of bioinformatic information online, we still need to conduct further experiments for the external validation of the results. Second, the details of the mechanisms by which these genes (CAST, WNT, and Lgr5) induce GC carcinogenesis remain to be elucidated. However, we could use databases to make preliminary reports on these genes, to facilitate confidence in future novel GC carcinogenetic models. Third, we need to perform tissue-sample confirmation due to the potential for errors in tumor purification.

## 5. Conclusions

Our study explored CAST as a signature oncogene in GCs. The CAST gene in gastric carcinogenesis was found to be regulated by macrophages in our OS analyses. The details of the mechanism of CAST-gene-related GC formation require further investigation; the mechanism is probably associated with Lgr5-related pathways and WNT/β-catenin cellular signaling.

## Figures and Tables

**Figure 1 biomolecules-12-00670-f001:**
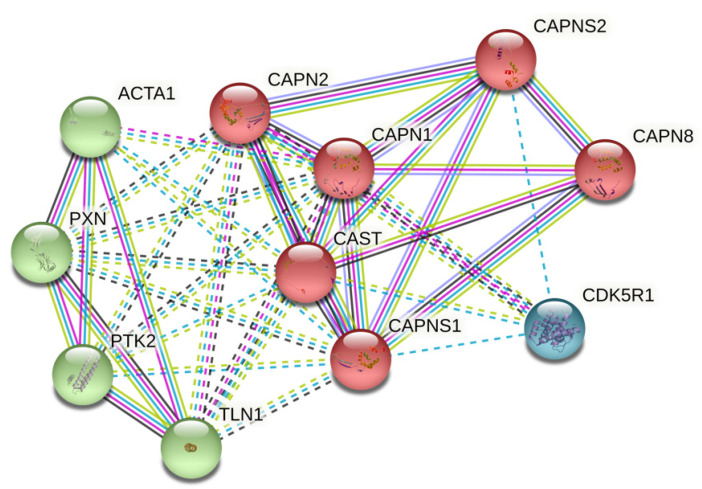
Important bio-targets of protein–protein interaction network in STRING clustering analysis network. Number of nodes: 11; number of edges: 40; average node degree: 7.27; average local clustering coefficient: 0.865; expected number of edges: 11; PPI enrichment *p*-value: 1.12 × 10^−11^.

**Figure 2 biomolecules-12-00670-f002:**
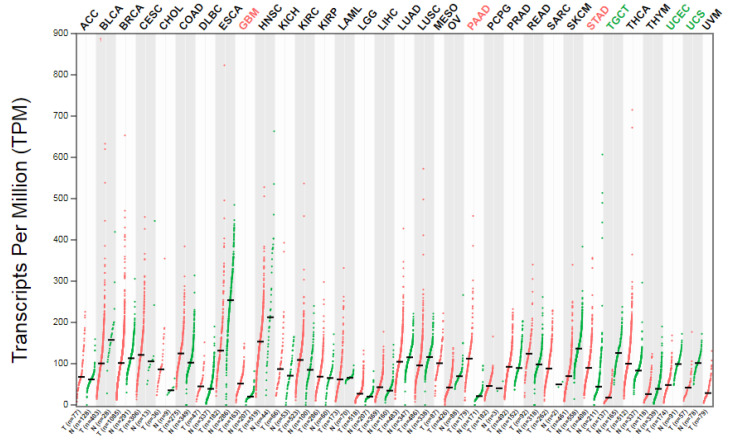
Bodymap indicating CAST expression; red for tumor and green for normal. Statistically significant in GBM, PAAD, STAD, TGCT, UCEC, and UCS.

**Figure 3 biomolecules-12-00670-f003:**
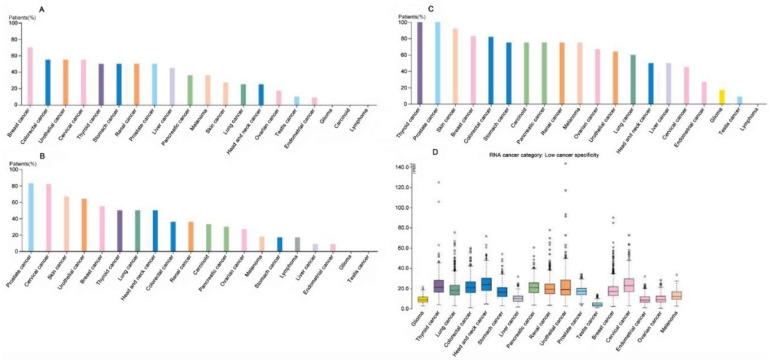
(**A**) HPA036881 protein expression in cancer tissues showing weak to moderate cytoplasmic immunoreactivity. Few cases of basal-cell carcinomas displayed strong immunoreactivity. Lymphomas along with several gliomas and testicular cancers were negative. (**B**) HPA036882 protein expression in cancer tissues displaying weak to moderate cytoplasmic staining with membranous positivity in several cases. Few cases of colorectal, breast, lung, skin, or urothelial cancers were strongly stained. Most cases of gliomas and testicular, liver, and endometrial cancers were negative. (**C**) CAB009491 protein expression in most cancer tissues demonstrated moderate to strong cytoplasmic positivity. Gliomas and lymphomas generally showed weak positivity, while testicular cancers, in most cases, were negative. (**D**) RNA-sequencing data for cancer category from the TCGA.

**Figure 4 biomolecules-12-00670-f004:**
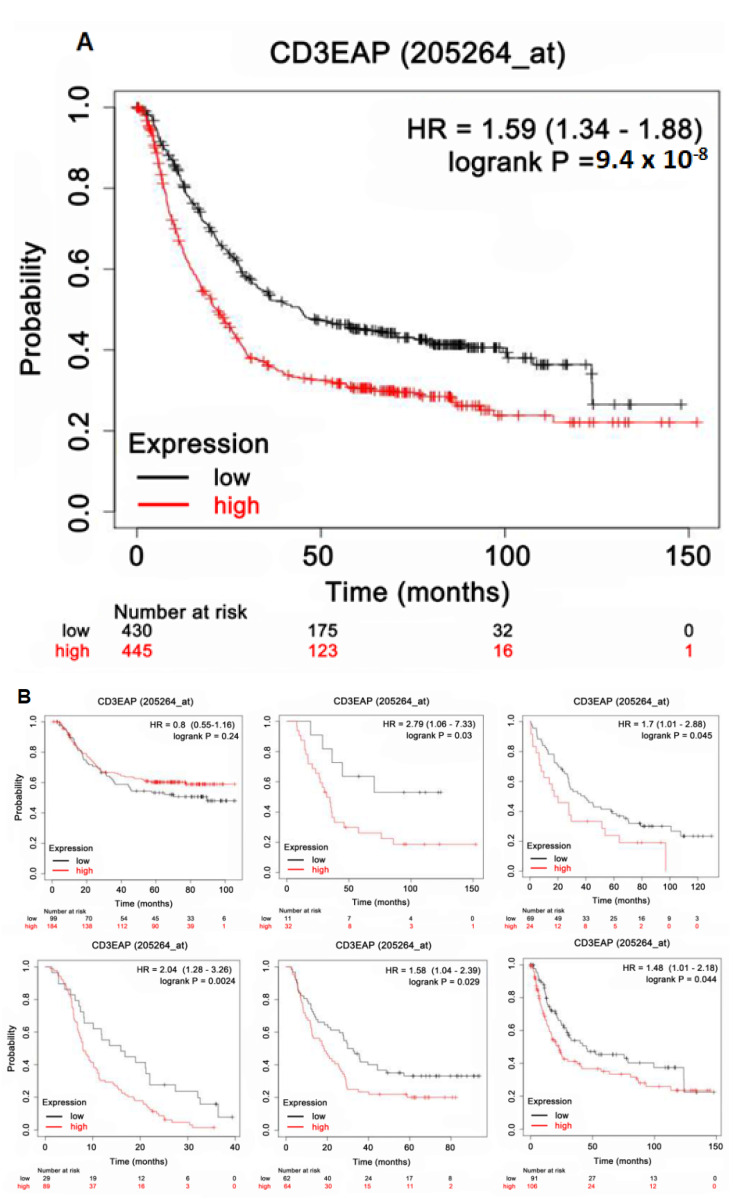
(**A**) All. High CAST expression versus low CAST expression in patients with GC. (**B**) Subgroup survival analyses. High CAST expression versus low CAST expression in patients with GC.

**Figure 5 biomolecules-12-00670-f005:**
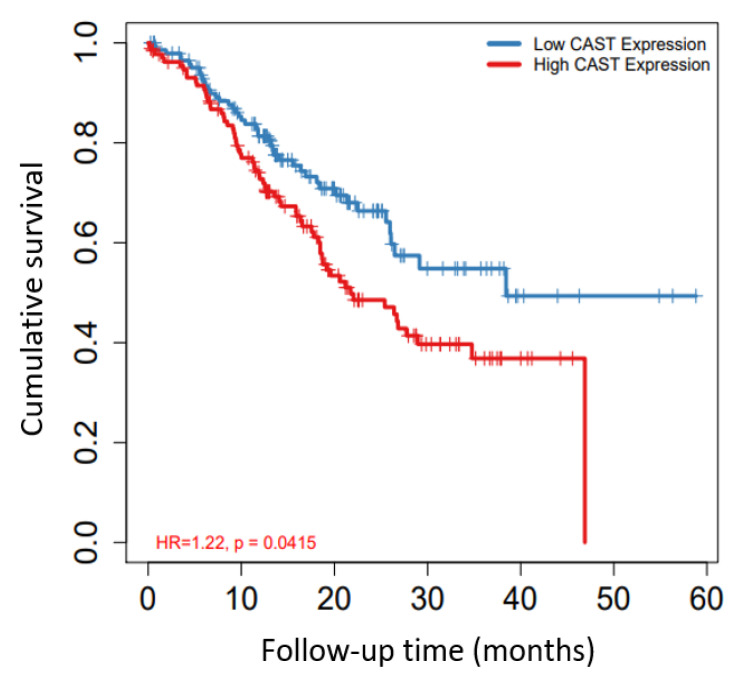
Cumulative survival for high and low CAST expression in GC. 5-year survival was evaluated.

**Figure 6 biomolecules-12-00670-f006:**
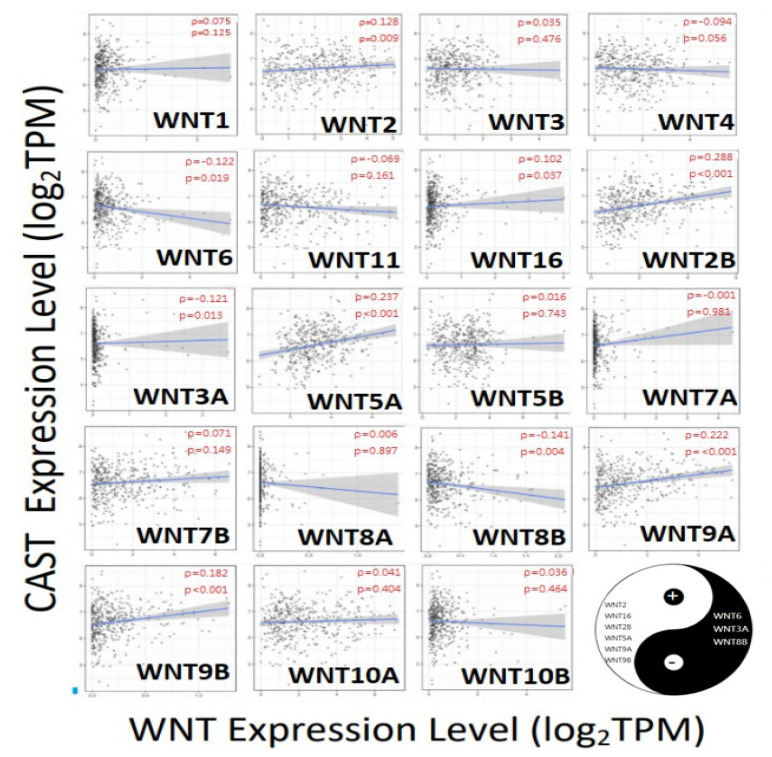
Correlation between CAST and the WNT family. +: significant positive correlation, WNT2 (rho = 0.128; *p*-value = 0.009); WNT16 (rho = 0.102; *p*-value = 0.037); WNT2B (rho = 0.288; *p*-value < 0.001); WNT5A (rho = 0.237; *p*-value < 0.001); WNT9A (rho = 0.222; *p*-value < 0.001); WNT9B (rho = 0.182; *p*-value < 0.001). −: significant negative correlation, WNT6 (rho = −0.122; *p*-value = 0.019); WNT3A (rho = −0.121; *p*-value = 0.013); WNT8B (rho = −0.141; *p*-value = 0.004).

**Figure 7 biomolecules-12-00670-f007:**
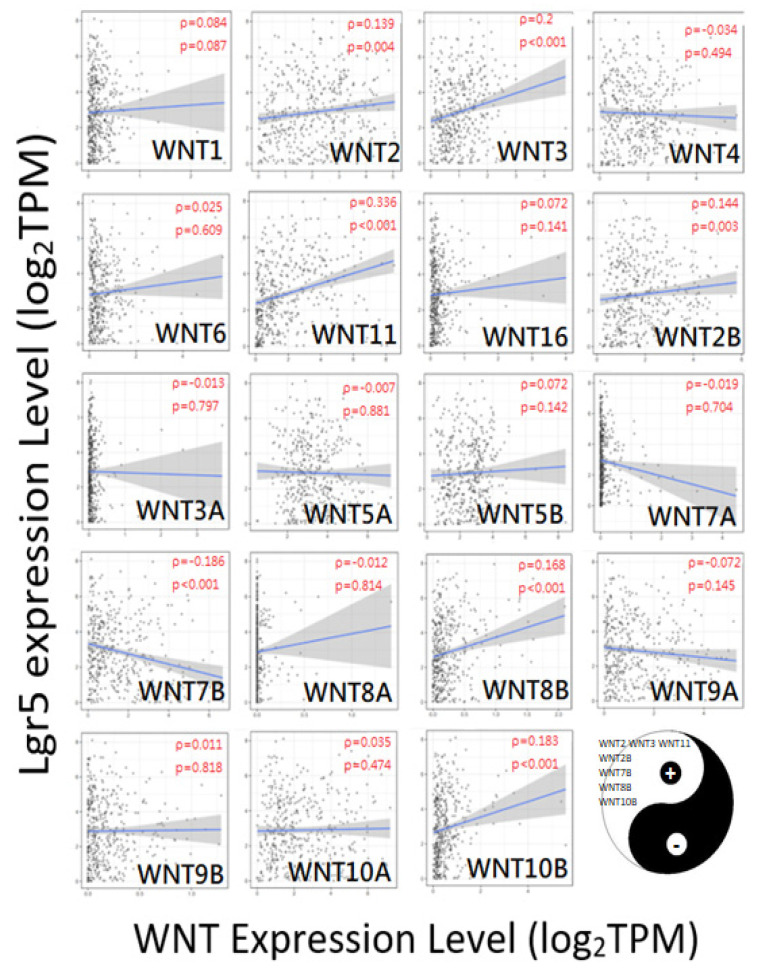
Correlation between CAST and the WNT family. +: significant positive correlation, WNT2 (rho = 0.139; *p*-value = 0.004); WNT3 (rho = 0.2; *p*-value < 0.001); WNT11 (rho = 0.336; *p*-value < 0.001); WNT2B (rho = 0.144; *p*-value = 0.003); WNT7B (rho = 0.186; *p*-value < 0.001); WNT8B (rho = 0.168; *p*-value < 0.001); WNT10B (rho = 0.183; *p*-value < 0.001). −: significant negative correlation, none.

**Figure 8 biomolecules-12-00670-f008:**
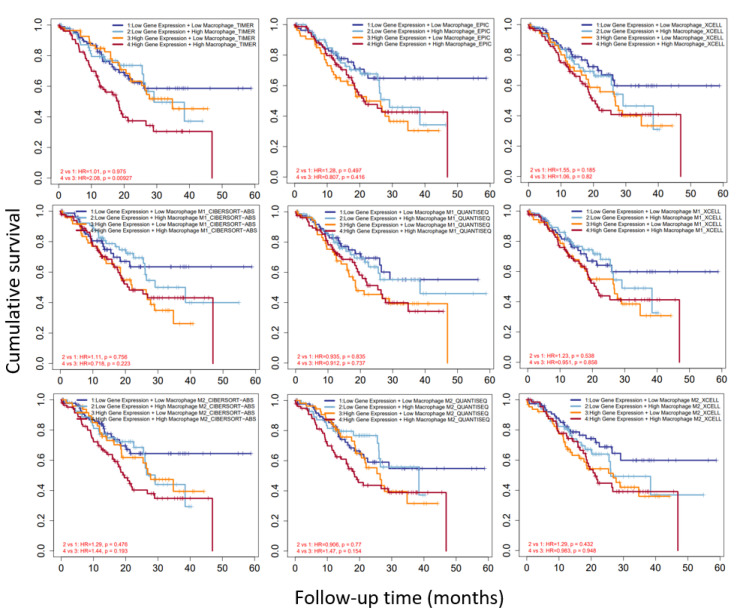
Cumulative survival in GC patients with CAST expression and macrophage infiltration. Statistically significant *p*-value in TIMER analyses.

**Figure 9 biomolecules-12-00670-f009:**
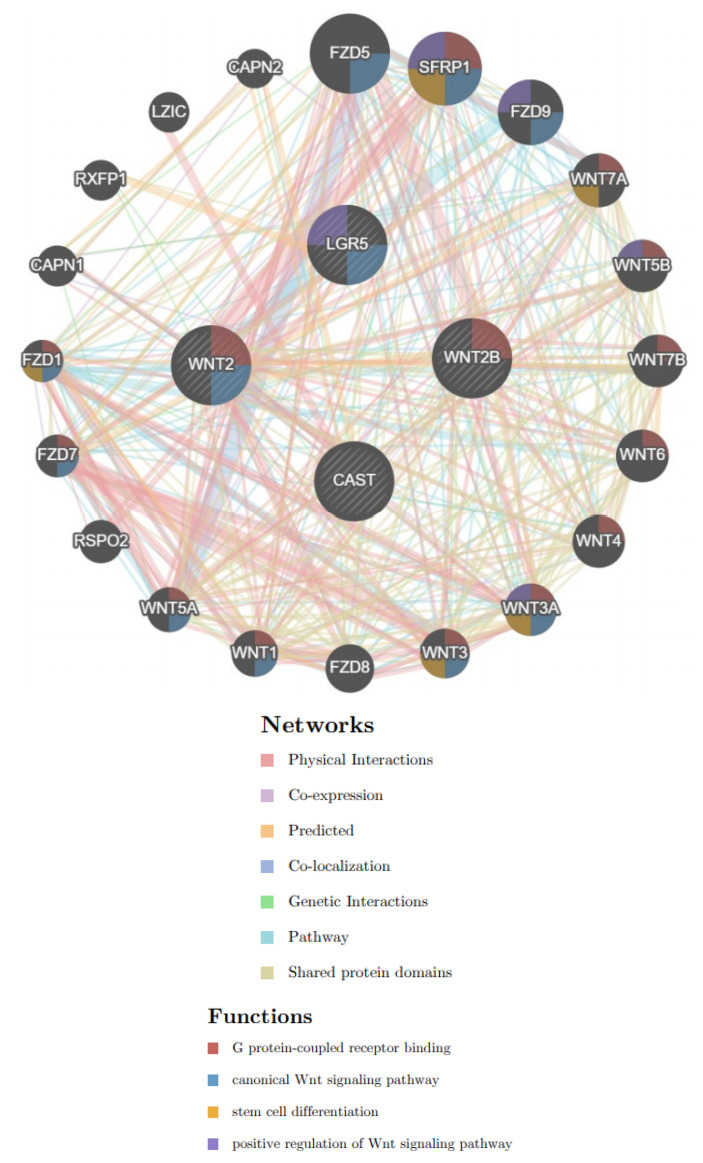
Interactions among CAST, WNT2/WNT2B, and Lgr5. Networks (physical interactions/co-expression/predicted/co-localization/genetic interactions/pathway/shared protein domains) and functions (G protein-coupled receptor binding/canonical Wnt signaling pathway/stem cell differentiation/positive regulation of Wnt signaling pathway) were involved.

**Table 1 biomolecules-12-00670-t001:** Gene ontology characteristics related to CAST.

GO-Term	Description	Count in Network	Strength	FDR
Biological Process				
GO:0022617	Extracellular matrix disassembly	4 of 66	2.03	0.00062
GO:0030198	Extracellular matrix organization	5 of 338	1.42	0.0044
GO:0007172	Signal complex assembly	2 of 8	2.65	0.0414
GO:1901699	Cellular response to nitrogen compound	5 of 645	1.14	0.0414
GO:0090130	Tissue migration	3 of 95	1.75	0.0417
GO:0016043	Cellular component organization	10 of 5447	0.51	0.0425
GO:0010506	Regulation of autophagy	4 of 340	1.32	0.0450
Molecular Function				
GO:0004198	Calcium-dependent cysteine-type endopeptidase activity	4 of 15	2.68	6.74 × 10^−7^
GO:0005509	Calcium-ion binding	6 of 703	1.18	0.00094
GO:0050839	Cell-adhesion-molecule binding	5 of 538	1.22	0.0037
GO:0008092	Cytoskeletal-protein binding	6 of 973	1.04	0.0037
GO:0017166	Vinculin binding	2 of 11	2.51	0.0103
GO:0005178	Integrin binding	3 of 147	1.56	0.0254
GO:0043167	Ion binding	10 of 6188	0.46	0.0255
Cellular Component				
GO:0005925	Focal adhesion	5 of 405	1.34	0.0028
GO:0001725	Stress fiber	3 of 65	1.91	0.0037
GO:0005829	Cytosol	10 of 5193	0.53	0.0037
GO:0031252	Cell leading edge	4 of 425	1.22	0.0125
GO:0015629	Actin cytoskeleton	4 of 477	1.17	0.0176
GO:0030027	Lamellipodium	3 of 202	1.42	0.0271

Abbreviations: GO, gene ontology; FRD, false discovery rate.

**Table 2 biomolecules-12-00670-t002:** The median expression of CAST in different tumor and normal samples.

Site	Tumor (TPM)	Normal (TPM)
Brain	51.76	20.16
Esophagus	131.69	253.86
Thyroid	99.97	83.24
Thymus	26.31	39.02
Blood	61.86	65.73
Lung	104.7	116.04
Breast	101.62	113.17
Liver	42.91	34.63
Biliary tract	86.27	35.47
Pancreas	112.23	21.35
Stomach	89.83	44.3
Adrenal gland	67.92	61.8
Kidney	109.01	85.14
Colon	124.76	102.46
Bladder	100.67	157.63
Prostate	92.38	89.49
Testis	17.75	125.93

**Table 3 biomolecules-12-00670-t003:** CAST RNA expression (according to sequencing) and survival analysis in GC cohorts.

Source	Median Survival		FDR	*p*-value
	Low Expression Cohort (Months)	High Expression Cohort (Months)		
All	44.57	21.93	1%	<0.0001
GSE62254	18.27	22.83	100%	0.2373
GSE22377	36.4	17.2	50%	0.0297
GSE51105	39.2	20.1	>50%	0.449
GSE14210	15.9	7.9	10%	0.0024
GSE29272	32.6	18.6	>50%	0.0289
GSE15459	45.1	22.8	>50%	0.0444

## Data Availability

The original contributions presented in the study are included in the article. Further inquiries can be directed to the first author (K.-T.Y.) and the corresponding author (J.-S.C.).

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
