# Peer review of "CAST as a Potential Oncogene, Identified by Machine Search, in Gastric Cancer Infiltrated with Macrophages and Associated with Lgr5"

_biomolecules, 2022, doi:10.3390/biom12050670_

Round 1

Reviewer 1 Report

The study conducted by authors on CAST as potential protein marker involved in GC is an interesting scientific question. However, after a robust lecture of the work, it can be improved well. in particular i suggest the following questions to be improved :

- in Figure 1. it can be necessary to describe well the network. in particular which means of edge linking node. Authors must be describe well the network analysis coming from string adding statistics and a new table reporting them. Also which pathways are involved? and which of them are most significant?

-The authors conducted in silico analyses by using different databases related both tissues and cell such as Protein atlas and GEPIA2. which are di differences in the expression between cell and tissues? what can be observed? authors must be include these informations.

 -in Paragraph 3.4 it can be necessary to describe well the obtained result about the OS. authors  added only figures without describe in deep way. must be improved. 

-in paragraph 3.6 authors   mention different databases. what contain that databases? which information can be retrieved? describe well this part.

Author Response

The study conducted by authors on CAST as potential protein marker involved in GC is an interesting scientific question.

However, after a robust lecture of the work, it can be improved well. in particular i suggest the following questions to be improved :

→ 

Thank you for considering our manuscript for further revisions.

(Pages and lines were in accordance with the revised manuscript with highlights.)

in Figure 1. it can be necessary to describe well the network. in particular which means of edge linking node. Authors must be describe well the network analysis coming from string adding statistics and a new table reporting them. Also which pathways are involved? and which of them are most significant?

→ 

Thanks for your valuable suggestions.

-In the the network, nodes are proteins, and the edges represent the predicted functional associations. The edges are drawn according to the view settings.

-Yes, we had added biostatistics in a table format. Please see the revised manuscript, Table 1.

(Page 5, line 183-184)

-We applied CAST (calpastatin) into STRING for proteiomics surveyance. Among the multiple pathways, we found the most significant one might be Calcium-dependent cysteine-type endopeptidase activity, with the highest strength and the lowest false discovery rate (FDR).

-The authors conducted in silico analyses by using different databases related both tissues and cell such as Protein atlas and GEPIA2. which are di differences in the expression between cell and tissues? what can be observed? authors must be include these informations.
→Yes, thanks for your comments. From Figrue 2 and Table 2, we could acknowledge that the significant CAST expression difference were present in esophagus, stomach, and pancreas tumor parts in contrast to normal parts. However, in Figure 3, we found CAST expression was not consistent. There was somehow individual’s genomic difference existed.

-in Paragraph 3.4 it can be necessary to describe well the obtained result about the OS. authors  added only figures without describe in deep way. must be improved. 
→ 

Yes, thanks for your suggestions. We had added further descriptions in paragraph 3.4. Please see them.

(Page 7, line 237-240)

-in paragraph 3.6 authors   mention different databases. what contain that databases? which information can be retrieved? describe well this part.

→Yes, thanks for your comments and we would describe this part. TIMER2.0 (http://timer.cistrome.org) is a comprehensive web resource for the systematic analysis of tumor-infiltrating immune cells across diverse cancers (Li T. et al., 2020). This giant and concrete database included immune infiltrates' abundances estimated by TIMER, EPIC, XCELL, CIBERSORT, QUANTISEQ…etc. Differential expression between tumor and adjacent normal tissues were saved across all The Cancer Genome Atlas (TCGA) tumors. Distributions of gene expression levels were displayed using box plots. 

Reviewer 2 Report

1-The character of macrophage infiltration in the the GC microenvironment also remains remains unaddressed. Please consider omitting the repeated words.

2- We first applied machine searching to evaluate gene candidates for GC. gene candidates with what characteristics?

3- Please explain in the abstract briefly why Lgr5 was tested? and also 

the machine-assisted search. 

4- Please avoid using acronyms in the introduction without introducing them first, for instance, Calpastatin (CAST), can appear this way.

5- We used Python Selenium (Version 3.8) to automatically search the TCGA database by entering different gene candidates, and we recorded all the candidate genes associated with the overall survival (OS) rate for GC. Then, the most relevant genes were precisely selected, including CAST and WNT (p-value < 0.001). Why was it necessary to look at various candidates if the objective was looking at CAST?

6- As a general note, all the figures require more extensive figure legend, not only a title but sufficient detail about the content of the figure. 

7- The abstract and objective section should point out that the expression of CAST was investigated pan-cancer, as shown in figure 2.

8- The raw TPM data obtained from each tumour-normal combination means very little without a statistical test. This also applies to figure 3A-C. please provide statistical tests for the protein expression levels in this figure. Also, the in-figure writing is very small making it difficult to understand the content.

9- In table 2, it is not clear if the p and q values are describing the log-rank test or that they may be unpaired student t-test assigning differences to low and high-risk groups.

10- Please define what hazard ratio refers to, for instance, the positive HR= 1.59 (1.34- 1.88) conveys increased risk compared to the reference group.

11- I have not found a subgroup survival analysis described in the main text to match figure 4B. In general, the findings have been reported in passing and they require much more in-depth explanation.

12- "Figure 5 shows the relevance of CAST alone to GC survival (HR) and the clinical outcome (HR: 1.22; p = 0.0415)". This is an example of not providing a narrative and starting a subsection by reporting the content of a figure. Please provide a narrative and through that refer to your figures. Line 246-266, are a strange way of reporting a Spearman Rho correlation, I would suggest incorporating these in sentences and also in your figure legend.

13- All the subsequent figures also suffer from a very little description, lack of sufficient introduction and explanation, and what the findings mean.

14- Table titles appear above the table, while figure legends should appear below figures. The data presented in figure 8 require a huge amount of explanation since there is so much data in it but the authors have given a highlight.

15- We discovered that, as shown in Figure 8, TIMER indicated that high CAST expression (>50%) and high macrophage infiltration (>50%) were significantly associated with lower cumulative survival than high CAST expression and low macrophage infiltration (<50%), with an HR of 2.08 and a p-value of 0.00927. What does CAST expression more than 50%> mean? What was the HR level calculated for it? Figure 9 needs more explanation.

Author Response

1-The character of macrophage infiltration in the the GC microenvironment also remains remains unaddressed. Please consider omitting the repeated words.
→ Yes, thanks for your advice. We will omit the repeated words.

2- We first applied machine searching to evaluate gene candidates for GC. gene candidates with what characteristics?
→ Yes, those gene candidates selected under machine search have significant GC-related survival characteristics in statistics.

3- Please explain in the abstract briefly why Lgr5 was tested? and also the machine-assisted search.
→ 

Yes, thanks for your comment. Lgr5 was tested due to its promising feature of intestinal stem cell upstream gene, with previous references confirmed.

Ref:

1.Fatehullah A, Terakado Y, Sagiraju S, et al. A tumour-resident Lgr5+ stem-cell-like pool drives the establishment and progression of advanced gastric cancers. Nat Cell Biol. 2021;23(12):1299-1313. doi:10.1038/s41556-021-00793-9

2.Wang X, Wang X, Liu Y, et al. LGR5 regulates gastric adenocarcinoma cell proliferation and invasion via activating Wnt signaling pathway [published correction appears in Oncogenesis. 2019 Feb 2;8(2):10]. Oncogenesis. 2018;7(8):57. Published 2018 Aug 9. doi:10.1038/s41389-018-0071-5

3.Wang Z, Liu C. Lgr5-Positive Cells are Cancer-Stem-Cell-Like Cells in Gastric Cancer. Cell Physiol Biochem. 2015;36(6):2447-2455. doi:10.1159/000430205

We applied the machine-assited search because of its convenience for surveying appropriate gene signature related to GC, Lgr5, and immune infiltrating tumor microenvironment, which is potential as a fresh promising marker in prevention and treatment strategy.

4- Please avoid using acronyms in the introduction without introducing them first, for instance, Calpastatin (CAST), can appear this way.

→ Yes, we will correct the mentioned errors.

(Page 2, line 84-85)

5- We used Python Selenium (Version 3.8) to automatically search the TCGA database by entering different gene candidates, and we recorded all the candidate genes associated with the overall survival (OS) rate for GC. Then, the most relevant genes were precisely selected, including CAST and WNT (p-value < 0.001). Why was it necessary to look at various candidates if the objective was looking at CAST?
→ 

Yes, thanks for your comments.

Our study group first selected several potential oncogenes of GC. After literature review and database mining, we finally chose CAST as our main research key role.

6- As a general note, all the figures require more extensive figure legend, not only a title but sufficient detail about the content of the figure. 
→ Yes, thanks for your comments and suggestions. We had extended our every figure legend in this article.

7- The abstract and objective section should point out that the expression of CAST was investigated pan-cancer, as shown in figure 2.
→ 

Thanks for your valuable comments. We had modified this error.

(Page 1, line 33-34)

8- The raw TPM data obtained from each tumour-normal combination means very little without a statistical test. This also applies to figure 3A-C. please provide statistical tests for the protein expression levels in this figure. Also, the in-figure writing is very small making it difficult to understand the content. 
→ 

Thanks for your valuable comments and suggestions. We had added the statistical tests for the protein expression levels to help understand the bioinformatic meaning and remove the obstacles of reading.

(Supplementary Table).

9- In table 2, it is not clear if the p and q values are describing the log-rank test or that they may be unpaired student t-test assigning differences to low and high-risk groups.
→ Yes, we identify your statement. Here we presented data retrieved from KM plotter to demonstrate that low CAST expression is associated with clinically significant better survival than high CAST expression.

10- Please define what hazard ratio refers to, for instance, the positive HR= 1.59 (1.34- 1.88) conveys increased risk compared to the reference group.
→ Yes, thanks for your comments and suggestions. Hazard ratio (HR) is a statistical index to describe the ratio of the hazard rates corresponding to the conditions described by two levels of an explanatory variable. Here we used HR to represent the survival risk along with time progression in GC patents on account of CAST expression.

11- I have not found a subgroup survival analysis described in the main text to match figure 4B. In general, the findings have been reported in passing and they require much more in-depth explanation.
→ 

Yes, thanks for the comments and suggestions. We had added in-depth explanations to match Figure 4B.

(Page 7, line 246-248)

12- "Figure 5 shows the relevance of CAST alone to GC survival (HR) and the clinical outcome (HR: 1.22; p = 0.0415)". This is an example of not providing a narrative and starting a subsection by reporting the content of a figure. Please provide a narrative and through that refer to your figures. Line 246-266, are a strange way of reporting a Spearman Rho correlation, I would suggest incorporating these in sentences and also in your figure legend.
→ 

Thanks for the comments and suggestions. We had modified our sentences and figure legends.

(Page 9, line 277-304)

13- All the subsequent figures also suffer from a very little description, lack of sufficient introduction and explanation, and what the findings mean.
→ Thanks for your comments. We had modified these shortcomings and supplied with adequate information for every figure.

14- Table titles appear above the table, while figure legends should appear below figures. The data presented in figure 8 require a huge amount of explanation since there is so much data in it but the authors have given a highlight.
→ Yes, we had modified table titles shown above the table and figure legends below figures. We had explained the Figure 8 content in paragraph 3.7 (Page 11, line 418-426).

15- We discovered that, as shown in Figure 8, TIMER indicated that high CAST expression (>50%) and high macrophage infiltration (>50%) were significantly associated with lower cumulative survival than high CAST expression and low macrophage infiltration (<50%), with an HR of 2.08 and a p-value of 0.00927. What does CAST expression more than 50%> mean? What was the HR level calculated for it? Figure 9 needs more explanation.
→ 

Yes, thanks for your suggestions and comments.

-High CAST expression indicated that one’s gene expression is more than 50% of the population. To avoid misunderstanding, we would remove the “50%”.

-The HR level was calculated for GC-related clinical survival.

-In Figure 9, we examined the current genomic network of CAST, WNT2, WNT2B, and Lgr5. Four functions were investigated, including G protein-coupled receptor binding, canonical Wnt signaling pathway, stem cell proliferation, and positive regulation of Wnt signaling pathway. The above-mentioned functions are strongly connected to our study topic and indicated future potential directions of GC treatments.

Round 2

Reviewer 1 Report

after revision of the manuscript, authors have improved it and they answered the questions posed.  they improved and worked on the manuscript as evidenced by the corrections.

Author Response

Thanks! 

Reviewer 2 Report

The authors have addressed my comments.

This manuscript is a resubmission of an earlier submission. The following is a list of the peer review reports and author responses from that submission.

Author Response

Thanks!